# Human NK cell development in hIL-7 and hIL-15 knockin NOD/SCID/IL2rgKO mice

Masashi Matsuda[1], Rintaro Ono[2], Tomonori Iyoda[3], Takaho Endo[4], Makoto Iwasaki[2], Mariko Tomizawa-Murasawa[2], Yoriko Saito[2], Akiko Kaneko[2], Kanako Shimizu[3], Daisuke Yamada[1], Narumi Ogonuki[5], Takashi Watanabe[4], Manabu Nakayama[6], Yoko Koseki[1], Fuyuko Kezuka-Shiotani[1], Takanori Hasegawa[1], Hiromasa Yabe[7], Shunichi Kato[7], Atsuo Ogura[5], Leonard D Shultz[8], Osamu Ohara[4,6] , Masaru Taniguchi[9], Haruhiko Koseki[1] , Shin-ichiro Fujii[3], Fumihiko Ishikawa[2]

The immune system encompasses acquired and innate immunity that matures through interaction with microenvironmental components. Cytokines serve as environmental factors that foster functional maturation of immune cells. Although NOD/SCID/IL2rgKO (NSG) humanized mice support investigation of human immunity in vivo, a species barrier between human immune cells and the mouse microenvironment limits human acquired as well as innate immune function. To study the roles of human cytokines in human acquired and innate immune cell development, we created NSG mice expressing hIL-7 and hIL-15. Although hIL-7 alone was not sufficient for supporting human NK cell development in vivo, increased frequencies of human NK cells were confirmed in multiple organs of hIL-7 and hIL-15 double knockin (hIL-7xhIL-15 KI) NSG mice engrafted with human hematopoietic stem cells. hIL-7xhIL-15 KI NSG humanized mice provide a valuable in vivo model to investigate development and function of human NK cells.

## Introduction

Cytokine receptor signaling is indispensable for reconstitution of the human immune system following hematopoietic stem cell (HSC) therapy. Among multiple cytokines, IL-7 promotes differentiation and maturation of T cells, B cells (Mackall et al, 2011), and innate lymphoid cells (Moro et al, 2010). In addition to the development of mature lymphoid cells, IL-7 signaling plays a pivotal role at the level of progenitor cells. Studies of IL-7– or IL-7R–deficient mice revealed multiple defects in T- and B-cell development (Peschon et al, 1994; von Freeden-Jeffry et al, 1995). Defective IL-7R expression in humans results in $T^-B^+NK^+$ SCID (Puel et al, 1998).

IL-15 supports innate lymphoid cell development (Ali et al, 2015). Studies using IL-15 transgenic mice (Fehniger et al, 2001) and IL-15 knockout (IL-15KO) mice (Kennedy et al, 2000) have shown IL-15 to be essential in the development of NK cells, natural killer T (NKT) cells, and memory CD8[+] T cells. Knocking out the genes encoding IL-15 or IL-15Rα results in complete loss of NK cells in the thymus, BM, and spleen. NKT cells and CD44[high] memory phenotype CD8[+] T cells were also reduced in IL-15KO and IL-15Rα knockout mice (Lodolce et al, 1998; Kennedy et al, 2000). A recent report demonstrated a role of IL-15 in anticancer immunity in that the frequencies of breast cancer metastasis were more frequent in IL-15KO mice than those in IL-15 transgenic mice or in C57BL/6 control mice (Gillgrass et al, 2014).

We developed NOD/SCID/IL2rgKO (NSG) mice to investigate the in vivo dynamics of the human immune system (Ishikawa et al, 2005; Shultz et al, 2005). In studies of humanized mice engrafted with human HSC, we and others reported development of human T and B cells. However, the frequencies of human NK cells did not reach physiological levels in NSG humanized mice (Andre et al, 2010). The decreased NK cell development could be due to the species barrier between human lymphoid or NK cell progenitors and recipient microenvironment (Mestas & Hughes, 2004).

To investigate the in vivo function of human IL-7 and IL-15 in the development of the human immune system, we created new strains of NSG mice expressing either hIL-7 alone (hIL-7TG NSG mice and hIL-7 KI NSG mice) and mice expressing hIL-7 and hIL-15 (hIL-7xhIL-15 KI NSG mice). Analyses of these mice engrafted with human HSCs showed that hIL-15 is required for NK cell development. In addition,

[1]Laboratory for Developmental Genetics, RIKEN Center for Integrative Medical Sciences, Yokohama, Japan   [2]Laboratory for Human Disease Models, RIKEN Center for Integrative Medical Sciences, Yokohama, Japan   [3]Laboratory for Immunotherapy, RIKEN Center for Integrative Medical Sciences, Yokohama, Japan   [4]Laboratory for Integrative Genomics, RIKEN Center for Integrative Medical Sciences, Yokohama, Japan   [5]Bioresource Engineering Division, RIKEN BioResource Center, Tsukuba, Japan   [6]Department of Technology Development, Kazusa DNA Research Institute, Kisarazu, Japan   [7]Department of Cell Transplantation and Regenerative Medicine, Tokai University School of Medicine, Isehara, Japan   [8]The Jackson Laboratory, Bar Harbor, ME, USA   [9]Laboratory for Immune Regulation, RIKEN Center for Integrative Medical Sciences, Yokohama, Japan

Correspondence: haruhiko.koseki@riken.jp; shin-ichiro.fujii@riken.jp; fumihiko.ishikawa@riken.jp

we found multiple subsets of human T cells in NSG recipient mice expressing human IL-7 and IL-15, demonstrating the roles of these cytokines in human T-cell development. These new humanized mouse models may support studies of human monoclonal antibody therapy in vivo and for studies of human acquired and innate tumor immunity.

# Results

## Reconstitution of human immunity in the presence of hIL-7

To study potential roles of human IL-7 in lymphoid cell development, we created hIL-7 KI and hIL-7 TG NSG mice. We first looked at effects of transgenic expression of human IL-7. When we compared reconstitution of T cells, B cells, and NK cells in the BM and spleen of cord blood (CB) HSC-engrafted NSG mice with or without expression of hIL-7, we did not find significant differences in the frequencies of each lineage within hCD45$^+$ cells (NSG, $n$ = 21: BM T cells 37.7 ± 5.7%, BM B cells 35.4 ± 3.8%, BM NK cells 1.0 ± 0.2%, spleen T cells 48.1 ± 4.8%, spleen B cells 44.6 ± 4.3%, spleen NK cells 0.7 ± 0.1%; hIL-7 TG NSG, $n$ = 3: BM T cells 28.7 ± 27.1%, BM B cells 42.0 ± 18.9%, BM NK cells 0.8 ± 0.2%, spleen T cells 38.5 ± 23.5%, spleen B cells 47.9 ± 18.8%, spleen NK cells 0.7 ± 0.3%; hIL-7 KI NSG, $n$ = 4: BM T cells 11.4 ± 5.9%, BM B cells 29.8 ± 8.9%, BM NK cells 1.2 ± 0.5%, spleen T cells 53.4 ± 11.3%, spleen B cells 30.9 ± 8.0%, spleen NK cells 2.6 ± 1.0%; representative flow cytometry plots shown in Fig S1A and summarized data shown in Table S1). Because IL-7 has been reported to promote B-cell differentiation and maturation, we next evaluated human B-cell subsets in hIL-7 KI NSG humanized mice. In the BM, the majority of human CD19$^+$ cells were CD10$^+$CD20$^-$ pro-B cells, while IgM$^+$IgD$^{low}$ transitional and IgM$^+$IgD$^+$ or IgM$^{low}$IgD$^+$ mature B cells accounted for more than 70% of human B cells in the spleen (representative flow cytometry plots shown in Fig S1B). The frequencies of human B-cell subsets were not altered significantly in hIL-7 KI NSG humanized mice (NSG, $n$ = 4: BM CD10$^+$CD20$^-$ cells 84.4 ± 3.9%, BM IgM$^+$ cells 23.7 ± 2.6%, spleen CD10$^+$CD20$^-$ cells 31.3 ± 6.3%, spleen IgM$^+$ cells 71.0 ± 5.4%; hIL-7 KI, $n$ = 3: BM CD10$^+$CD20$^-$ cells 81.9 ± 1.9%, BM IgM$^+$ cells 23.3 ± 5.9%, spleen CD10$^+$CD20$^-$ cells 21.0 ± 5.4%, spleen IgM$^+$ cells 73.4 ± 2.1%; summarized data shown in Table S2).

## Reconstitution of human immunity in the presence of hIL-7 and hIL-15

To examine the role of hIL-15 and hIL-7 in human immune cell development, a new strain of NSG mice expressing hIL-15 and both hIL-7 and hIL-15 (hIL-7xhIL-15 KI NSG) were created. Plasma concentrations of human IL-7 and human IL-15 in hIL-7xhIL-15 KI NSG mice ($n$ = 8) were 1.9 ± 0.3 pg/ml and 87.8 ± 9.8 pg/ml, respectively (Fig S2). We then reconstituted human immunity by intravenously injecting 2.5 × 10$^3$ to 2.6 × 10$^4$ CB CD34$^+$CD38$^-$CD45RA$^-$ or CD34$^+$CD38$^-$CD90$^+$CD45RA$^-$ HSCs into NSG or hIL-7xhIL-15 KI NSG newborns (Table 1). In the recipient BM, thymus, spleen, and peripheral blood (PB), we found significantly higher frequencies of human CD56$^+$ NK cells in hIL-7xhIL-15 KI NSG humanized mice

compared with conventional NSG humanized mice (NSG, $n$ = 21: BM NK cells 1.0 ± 0.2%, spleen NK cells 0.7 ± 0.1%, PB NK cells 1.0 ± 0.2%; hIL-7xhIL-15 KI NSG, $n$ = 17: BM NK cells 17.7 ± 4.7%, spleen NK cells 28.3 ± 5.3%, PB NK cells 43.1 ± 5.4%; NSG, $n$ = 18: thymus NK cells 0.8 ± 0.2%; and hIL-7xhIL-15 KI NSG, $n$ = 13: thymus NK cells 7.2 ± 1.4%. $P$ < 0.001 in BM, spleen, PB, and thymus, by two-tailed $t$ test; Fig 1A–C, absolute numbers of cells are shown in Table S3). We confirmed efficient differentiation of human CD56$^+$ NK cells in the BM, spleen, and PB of hIL-15 KI NSG humanized mice (representative flow cytometry plots shown in Fig S3 and summarized data shown in Table S4); therefore, we expected the presence of both hIL-7 and hIL-15 to further enhance development of human innate lymphoid cells.

hIL-15 has been reported to be secreted from myeloid cells (Cui et al, 2014) by trans-presentation (Stonier & Schluns, 2010). Consistent with this, we confirmed expression of human IL-15 in Gra1$^+$ Mac1$^+$ mouse myeloid cells of hIL-7xhIL-15 KI NSG humanized mice, not in CD45$^+$CD33$^+$ human myeloid cells or Gra1$^+$Mac1$^+$ mouse myeloid cells purified from conventional NSG humanized mice (NSG $n$ = 3 and IL7xIL15 $n$ = 3; Fig 1D). In addition, we detected tissue-resident NK cells such as CXCR6$^+$ NK cells (Stegmann et al, 2016) in BM and liver of hIL-7xhIL-15 KI NSG humanized mice (BM $n$ = 8, Liver $n$ = 5, PB $n$ = 4: BM CXCR6$^+$CD56$^+$NK cells 67.6 ± 2.3%, liver CXCR6$^+$CD56$^+$ NK cells 10.2 ± 5.1%, PB CXCR6$^+$CD56$^+$NK cells 1.4 ± 0.9%; Fig 1E and F).

In addition to human NK cell development, we examined whether the cytokines facilitate the development of human T cells or induce skewing of specific human T-cell subsets. To this end, we first analyzed development of human T-cell subsets in the thymus and spleen. Within the thymic hCD45$^+$CD56$^-$CD3$^+$ fraction, we found higher frequency of CD4$^-$CD8$^+$ single positive T cells in hIL-7xhIL-15 KI NSG humanized mice (NSG, $n$ = 18: CD4$^-$CD8$^+$ cells 15.6 ± 3.1%; hIL7xhIL-15 KI NSG, $n$ = 12: CD4$^-$CD8$^+$ cells 59.5 ± 5.9%. $P$ < 0.001 by two-tailed $t$ test; Fig 2A and B, absolute numbers of cells are shown in Table S3).

Next, using spleen cells, we examined the frequencies of human CD3$^+$ T, CD4$^+$ T, and CD8$^+$ T cells and their CD45RA and Foxp3 expression in NSG and hIL-7xhIL-15 KI NSG humanized mice (representative flow cytometry plots shown in Fig 2C). We found that both CD45RA$^+$ naive and CD45RA$^-$ memory T cells were present in CD4$^+$ T or CD8$^+$ T-cell populations of hIL-7xhIL-15 KI NSG humanized mice (NSG, $n$ = 7: CD4$^+$CD45RA$^+$ cells 3.3 ± 1.0%, CD4$^+$CD45RA$^-$ cells 96.7 ± 1.0%, CD8$^+$CD45RA$^+$ cells 30.6 ± 6.6%, CD8$^+$CD45RA$^-$ cells 69.5 ± 6.6%; hIL-7xhIL-15 KI NSG, $n$ = 5: CD4$^+$CD45RA$^+$ cells 11.3 ± 6.7%, CD4$^+$CD45RA$^-$ cells 88.7 ± 6.8%, CD8$^+$CD45RA$^+$ cells 37.0 ± 17.2%, CD8$^+$ CD45RA$^-$ cells 63.0 ± 17.2%; Fig 2D). Frequencies of splenic CD4$^+$ and CD8$^+$ T cells in CD3$^+$ T cells are shown in Fig 2D (NSG, $n$ = 7: CD4$^+$ cells 72.8 ± 3.1%, CD8$^+$ cells 21.5 ± 3.0%; hIL-7xhIL-15 KI NSG, $n$ = 5: CD4$^+$ cells 61.9 ± 4.7%, CD8$^+$ cells 31.6 ± 3.5%. $P$ = 0.068 in CD4$^+$ cells and $P$ = 0.052 in CD8$^+$ cells by two-tailed $t$ test; Fig 2D). Frequency of Foxp3$^+$ cells in CD4$^+$ T cells was similar between NSG and hIL-7xhIL-15 KI NSG humanized mice (NSG, $n$ = 7: 8.8 ± 1.5%; hIL-7xhIL-15 KI NSG, $n$ = 5: 6.5 ± 2.4%. $P$ = 0.41 by two-tailed $t$ test). Although we found enhanced human NK cells and T-cell development in hIL-7xhIL-15 KI NSG humanized mice, TCRV$\alpha$24$^+$V$\beta$11$^+$CD3$^+$ NKT cells were not detected (representative flow cytometry plots in Fig 2E). This suggests that hIL-7 and hIL-15 play distinct roles in the development of human NK and NKT cells.

**Table 1.  Engraftment of human leukocytes in NSG mice expressing hIL-7 and/or hIL-15.**

| Mouse ID | CB ID | Strain | Week[a] | Spleen, % | | | | | PB, % | | | | |
|---|---|---|---|---|---|---|---|---|---|---|---|---|---|
| | | | | hCD45+[b] | CD56+ | CD3+ | CD19+ | CD33+ | hCD45+[b] | CD56+ | CD3+ | CD19+ | CD33+ |
| IL7x15#1 | CB#1 | hIL-7xhIL-15 KI | 15 | 46.6 | 35.3 | 11.4 | 19.2 | 15.3 | 15.0 | 60.4 | 6.5 | 2.4 | 6.1 |
| IL7x15#2 | CB#1 | hIL-7xhIL-15 KI | 18 | 35.8 | 38.6 | 1.3 | 11.1 | 10.4 | 30.5 | 68.5 | 1.3 | 11.1 | 10.4 |
| IL7x15#3 | CB#2 | hIL-7xhIL-15 KI | 10 | 98.1 | 13.0 | 0.1 | 78.8 | 3.1 | 93.6 | 38.5 | 0.3 | 42.2 | 7.8 |
| IL7x15#4 | CB#2 | hIL-7xhIL-15 KI | 16 | 92.7 | 3.8 | 23.5 | 57.8 | 5.7 | 95.9 | 19.2 | 37.4 | 15.4 | 10.1 |
| IL7x15#5 | CB#3 | hIL-7xhIL-15 KI | 13 | 76.0 | 18.7 | 0.2 | 67.1 | 3.5 | 22.3 | 70.6 | 0.1 | 11.7 | 9.4 |
| IL7x15#6 | CB#4 | hIL-7xhIL-15 KI | 19 | 77.7 | 22.1 | 2.5 | 49.0 | 4.1 | 30.4 | 37.7 | 2.5 | 26.7 | 4.9 |
| IL7x15#7 | CB#4 | hIL-7xhIL-15 KI | 22 | 66.8 | 26.0 | 26.8 | 26.5 | 3.6 | 18.8 | 38.3 | 33.1 | 21.0 | 1.4 |
| IL7x15#8 | CB#4 | hIL-7xhIL-15 KI | 23 | 64.4 | 16.2 | 10.1 | 16.2 | 6.6 | 15.7 | 6.0 | 18.2 | 24.3 | 3.1 |
| IL7x15#9 | CB#5 | hIL-7xhIL-15 KI | 13 | 80.6 | 36.6 | 1.7 | 40.8 | 8.2 | 49.2 | 47.2 | 1.2 | 19.3 | 20.0 |
| IL7x15#10 | CB#6 | hIL-7xhIL-15 KI | 19 | 65.9 | 52.9 | 1.0 | 17.3 | 14.5 | 30.8 | 64.9 | 0.6 | 6.9 | 8.3 |
| IL7x15#11 | CB#7 | hIL-7xhIL-15 KI | 16 | 97.0 | 1.2 | 74.6 | 15.4 | 3.4 | 98.7 | 9.8 | 69.2 | 1.2 | 1.0 |
| IL7x15#12 | CB#7 | hIL-7xhIL-15 KI | 17 | 92.3 | 2.6 | 72.5 | 17.6 | 2.6 | 85.8 | 14.3 | 67.6 | 3.6 | 1.8 |
| IL7x15#13 | CB#8 | hIL-7xhIL-15 KI | 13 | 51.1 | 21.7 | 0.9 | 60.0 | 11.5 | 4.8 | 37.6 | 0.7 | 13.3 | 22.6 |
| IL7x15#14 | CB#9 | hIL-7xhIL-15 KI | 20 | 28.8 | 56.0 | 2.5 | 11.5 | 9.2 | 11.6 | 70.2 | 1.4 | 7.6 | 10.5 |
| IL7x15#15 | CB#10 | hIL-7xhIL-15 KI | 11 | 78.7 | 9.4 | 0.4 | 76.1 | 5.1 | 77.1 | 25.2 | 0.3 | 34.7 | 27.9 |
| IL7x15#16 | CB#11 | hIL-7xhIL-15 KI | 15 | 58.7 | 72.8 | 1.1 | 12.9 | 6.7 | 7.4 | 68.8 | 0.0 | 8.5 | 15.3 |
| IL7x15#17 | CB#12 | hIL-7xhIL-15 KI | 15 | 74.1 | 54.1 | 0.8 | 27.8 | 8.1 | 20.1 | 55.4 | 0.9 | 20.8 | 2.0 |
| IL7x15#18 | CB#13 | hIL-7xhIL-15 KI | 8 | 68.9 | 25.6 | 0.0 | 54.1 | 7.8 | 7.7 | 24.6 | 0.0 | 29.1 | 33.0 |
| IL7x15#19 | CB#14 | hIL-7xhIL-15 KI | 16 | 23.9 | 87.5 | 0.5 | 7.6 | 1.5 | 2.2 | 82.8 | 0.8 | 5.5 | 3.1 |
| IL7x15#20 | CB#15 | hIL-7xhIL-15 KI | 9 | 56.9 | 31.3 | 0.0 | 32.3 | 22.5 | 35.9 | 60.0 | 0.1 | 8.2 | 18.2 |
| IL7x15#21 | CB#16 | hIL-7xhIL-15 KI | 11 | 46.5 | 61.5 | 1.2 | 24.1 | 3.6 | 31.2 | 70.7 | 1.2 | 12.4 | 4.9 |
| IL7x15#22 | CB#17 | hIL-7xhIL-15 KI | 10 | 90.4 | 8.6 | 0.4 | 72.5 | 8.6 | 75.6 | 29.8 | 0.8 | 35.6 | 28.3 |
| IL7x15#23 | CB#17 | hIL-7xhIL-15 KI | 11 | 77.3 | 7.7 | 0.5 | 77.3 | 10.1 | 61.7 | 13.8 | 0.8 | 49.2 | 27.5 |
| IL7KI#1 | CB#13 | hIL-7 KI | 18 | 99.5 | 0.4 | 1.5 | 11.5 | 83.9 | 97.2 | 1.1 | 4.4 | 4.6 | 85.4 |
| IL7KI#2 | CB#13 | hIL-7 KI | 19 | 87.9 | 5.0 | 10.7 | 36.6 | 37.9 | 83.3 | 11.7 | 20.0 | 23.4 | 27.8 |
| IL7KI#3 | CB#14 | hIL-7 KI | 21 | 90.5 | 3.1 | 10.3 | 25.9 | 57.1 | 59.9 | 4.5 | 18.4 | 40.9 | 25.6 |
| IL7KI#4 | CB#14 | hIL-7 KI | 19 | 93.7 | 1.9 | 10.4 | 49.4 | 34.5 | 98.4 | 1.9 | 63.4 | 14.1 | 15.0 |
| IL7TG#5 | CB#15 | hIL-7 Tg | 16 | 97.8 | 1.2 | 8.0 | 58.6 | 22.2 | 92.3 | 1.7 | 32.1 | 25.2 | 26.2 |
| IL7TG#6 | CB#16 | hIL-7 Tg | 12 | 81.0 | 0.6 | 2.5 | 73.9 | 8.5 | 93.1 | 0.7 | 2.2 | 73.9 | 14.4 |
| IL7TG#7 | CB#17 | hIL-7 Tg | 24 | 95.6 | 0.3 | 2.1 | 11.3 | 84.9 | 81.9 | 0.3 | 0.2 | 2.8 | 95.4 |
| NSG#1 | CB#1 | NSG | 20 | 79.8 | 0.3 | 48.6 | 40.1 | 1.6 | 43.6 | 0.2 | 91.1 | 5.0 | 0.7 |
| NSG#2 | CB#2 | NSG | 17 | 78.1 | 0.2 | 77.5 | 19.1 | 1.4 | 68.5 | 0.2 | 96.9 | 1.0 | 0.2 |
| NSG#3 | CB#2 | NSG | 18 | 85.9 | 0.6 | 14.8 | 74.5 | 3.9 | 37.8 | 0.6 | 50.8 | 28.9 | 11.4 |
| NSG#4 | CB#3 | NSG | 18 | 90.3 | 1.2 | 7.5 | 76.7 | 6.4 | 59.8 | 1.2 | 15.2 | 51.6 | 17.2 |
| NSG#5 | CB#3 | NSG | 19 | 92.4 | 1.2 | 20.5 | 72.4 | 3.5 | 81.5 | 2.5 | 44.0 | 44.3 | 3.6 |
| NSG#6 | CB#3 | NSG | 20 | 91.7 | 1.1 | 10.8 | 72.6 | 5.4 | 68.1 | 1.0 | 28.2 | 36.7 | 22.4 |
| NSG#7 | CB#5 | NSG | 18 | 71.6 | 0.9 | 52.4 | 47.9 | 1.2 | 37.4 | 0.8 | 74.5 | 20.5 | 1.7 |
| NSG#8 | CB#5 | NSG | 19 | 90.2 | 0.2 | 77.6 | 24.0 | 0.9 | 79.9 | 0.6 | 89.2 | 7.0 | 2.3 |
| NSG#9 | CB#7 | NSG | 17 | 98.8 | 0.1 | 64.8 | 30.9 | 1.2 | 97.9 | 0.6 | 83.3 | 10.6 | 0.3 |
| NSG#10 | CB#8 | NSG | 19 | 88.6 | 2.3 | 40.9 | 48.2 | 6.4 | 36.1 | 1.9 | 46.1 | 42.9 | 4.9 |
| NSG#11 | CB#8 | NSG | 19 | 88.5 | 0.7 | 35.3 | 58.1 | 4.8 | 39.4 | 0.3 | 46.6 | 48.7 | 2.3 |
| NSG#12 | CB#10 | NSG | 20 | 98.6 | 0.1 | 60.2 | 32.1 | 0.8 | 97.0 | 0.5 | 85.5 | 7.7 | 1.3 |

**Table 1.  Continued**

| Mouse ID | CB ID | Strain | Week[a] | Spleen, % | | | | | PB, % | | | | |
|---|---|---|---|---|---|---|---|---|---|---|---|---|---|
| | | | | hCD45+[b] | CD56+ | CD3+ | CD19+ | CD33+ | hCD45+[b] | CD56+ | CD3+ | CD19+ | CD33+ |
| NSG#13 | CB#10 | NSG | 20 | 93.9 | 0.8 | 56.7 | 38.7 | 3.2 | 79.8 | 2.3 | 78.8 | 7.2 | 5.6 |
| NSG#14 | CB#12 | NSG | 19 | 84.4 | 1.0 | 37.8 | 55.8 | 3.3 | 56.7 | 0.9 | 54.1 | 41.7 | 1.6 |
| NSG#15 | CB#12 | NSG | 21 | 85.1 | 0.7 | 40.4 | 53.1 | 2.7 | 49.9 | 0.7 | 56.0 | 34.7 | 3.8 |
| NSG#16 | CB#13 | NSG | 22 | 95.9 | 0.2 | 75.9 | 13.4 | 2.2 | 97.4 | 0.4 | 90.0 | 2.2 | 3.6 |
| NSG#17 | CB#13 | NSG | 23 | 78.5 | 1.8 | 61.8 | 27.1 | 3.9 | 28.5 | 4.3 | 64.8 | 18.9 | 3.5 |
| NSG#18 | CB#14 | NSG | 21 | 82.1 | 0.4 | 38.8 | 56.4 | 2.3 | 31.5 | 0.2 | 63.3 | 29.1 | 3.4 |
| NSG#19 | CB#18 | NSG | 17 | 93.7 | 0.7 | 55.3 | 38.9 | 1.5 | 62.0 | 1.0 | 75.8 | 18.5 | 1.8 |
| NSG#20 | CB#18 | NSG | 19 | 83.3 | 0.6 | 49.7 | 43.9 | 0.9 | 66.1 | 0.6 | 88.9 | 6.3 | 0.1 |
| NSG#21 | CB#19 | NSG | 20 | 99.7 | 0.3 | 83.0 | 13.4 | 1.4 | 99.0 | 0.5 | 90.9 | 4.9 | 1.5 |
| NSG#22 | CB#18 | NSG | 19 | 89.9 | 1.7 | 54.5 | 36.5 | 2.1 | 64.7 | 1.4 | 76.5 | 18.1 | 1.4 |
| NSG#23 | CB#18 | NSG | 19 | 93.6 | 0.5 | 16.0 | 79.6 | 1.0 | 81.7 | 0.5 | 20.2 | 76.7 | 0.2 |
| NSG#24 | CB#19 | NSG | 17 | 87.0 | 2.0 | 46.4 | 45.1 | 3.3 | 42.8 | 7.0 | 65.9 | 14.1 | 2.8 |

[a]Number of weeks after CB transplantation when euthanized are shown.
[b]Frequencies of human CD45+ cells in total leukocytes including human and mouse CD45+ cells.

## Maturation and location of human NK cells in hIL-7xhIL-15 KI NSG humanized mice

After the development of human NK cells in the primary immune organs (BM and thymus), they undergo maturation in peripheral organs such as spleen (Freud et al, 2014; Bjorkstrom et al, 2016). We analyzed frequencies of human NK cells and assessed the degree of maturation in hIL7xhIL15 KI NSG humanized mice. Among CD56+ cells, we found significantly higher percentages of human CD56+CD94+CD16+ relatively mature NK cells in the spleen and PB of the hIL-7xhIL-15 KI NSG humanized mice compared with NSG humanized mice (NSG, $n$ = 20: BM CD94+CD16+ cells 28.8 ± 4.7%, spleen CD94+CD16+ cells 27.2 ± 2.8%; hIL-7xhIL-15 KI NSG, $n$ = 16: BM CD94+CD16+ cells 42.6 ± 4.4%, spleen CD94+CD16+ cells 73.6 ± 3.1%; NSG, $n$ = 17: PB CD94+CD16+ cells 40.7 ± 4.1%; hIL-7xhIL-15 KI NSG, $n$ = 13: PB CD94+CD16+ cells 83.8 ± 2.3%. $P$ < 0.001 in spleen and PB by two-tailed $t$ test; Fig 3A and B absolute numbers of cells are shown in Table S3). In line with phenotypical maturation, human NK cells developing in hIL-7xhIL-15 NSG humanized mice showed higher expression of perforin and granzyme B, CD56+ NK cell subset (NSG, $n$ = 13: perforin+ cells 25.1 ± 5.6%, granzyme B+ cells 17.8 ± 4.7%, perforin+ granzyme B+ cells 12.7 ± 3.9%; hIL-7xhIL-15 KI NSG, $n$ = 10: perforin+ cells 61.4 ± 4.3%, granzyme B+ cells 47.0 ± 4.3%, perforin+ granzyme B+ cells 39.5 ± 3.4%. $P$ < 0.001 in perforin+ cells, perforin+ granzyme B+ cells, and granzyme B+ cells by two-tailed $t$ test; Fig 3C and D, absolute numbers of cells are shown in Table S3). We next assessed functional capacity of human NK cells developing in conventional NSG humanized mice and hIL-7xhIL-15 NSG humanized mice using in vitro cytotoxicity assay. hIL-7xhIL-15 NSG spleen NK cells exhibited cytotoxicity at a similar level compared with human PB NK cells (NSG, $n$ = 3: 5.7% ± 2.2%; hIL-7xhIL-15 KI NSG, $n$ = 8: 12.4 ± 1.8%; human PB, $n$ = 2: 8.1 ± 1.8%; Fig 3E).

When we analyzed PFA-fixed, paraffin-embedded thin sections of the recipient spleens, we found cells that were specifically stained with anti-hCD56 and NKp46 antibody. Whereas hCD3+ T cell and hCD19+ B cells appeared to form lymphoid clusters, human NK cells were distributed outside lymphoid clusters, consistent with physiological distribution of human NK cells in human spleen (Witte et al, 1990) (Figs 4 and S4). These findings indicate that hIL-7 and hIL-15 support multiorgan development and maturation of human NK cells in humanized mice.

## Maintenance of human NKT cells in hIL-7xhIL-15 KI NSG mice

Although TCRVα24+Vβ11+CD3+ NKT cells were not detected in the BM and spleen of hIL-7xhIL-15 KI NSG humanized mice, we examined whether human IL-7 and human IL-15 maintain human NKT cells for longer term in the BM and lungs. We intravenously injected $2 × 10^6$ cells, human NKT cells, or iPS-NKT cells into NSG mice or hIL-7xhIL-15 KI NSG mice. In hIL-7xhIL-15 KI NSG mice, we detected NKT cells or iPS-NKT cells in the BM and lungs at higher frequencies at 14 d postinjection (NSG, $n$ = 4: BM NKT cells 0.000%, lung NKT cells 0.001 ± 0.002%; hIL-7xhIL-15 $n$ = 11 BM NKT cells 0.084 ± 0.025%, lung NKT cells 0.079 ± 0.014%, $P$ = 0.07 in BM and $P$ < 0.01 in the lungs by two-tailed $t$ test; Fig S5). This finding indicates that hIL-7 and hIL-15 play an important role for the survival of human NKT cells.

## Gene expression signatures of human NK cells in hIL-7xhIL-15 KI NSG mice

Furthermore, we examined gene expression profiles of human NK cells in hIL-7xhIL-15 KI NSG humanized mice by RNA sequencing (Fig 5). We purified hCD45+CD56+CD3− splenic NK cells from hIL-7xhIL-15 KI NSG and conventional NSG humanized mice. In comparison with gene expression signatures of human NK cells in conventional NSG humanized mice and human PBMC-derived NK cells (GSE64655) (Hoek et al, 2015), expression of *PRDM1, CCL4L1, CCL4, KIR2DS4, KIR2DL1, KIR2DL3, KIR3DL1, and KIR2DP1* was up-regulated in human NK cells from hIL-7xhIL-15 KI NSG humanized mice. In addition,

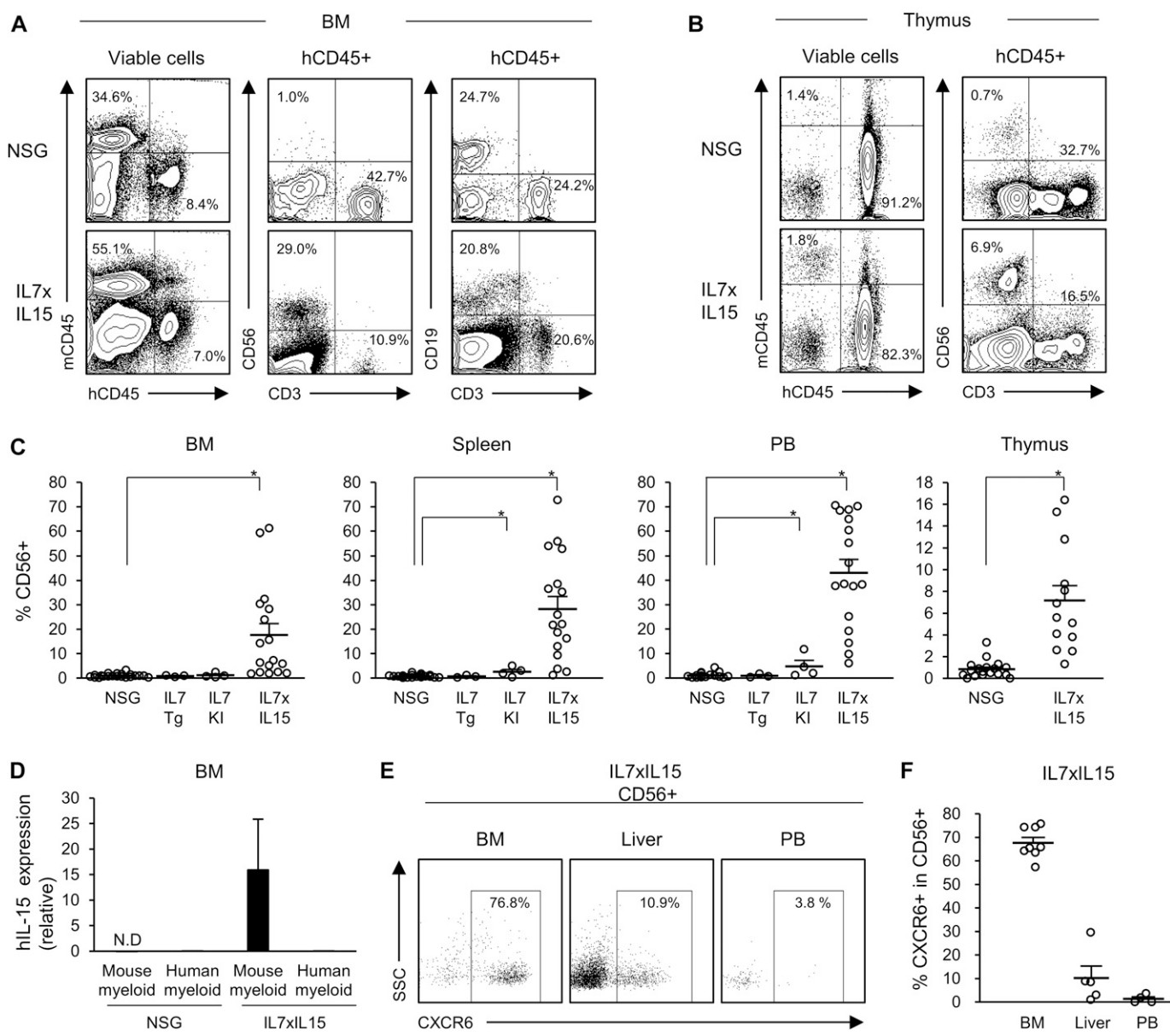

**Figure 1. Increased reconstitution of human NK cells in NSG hIL-7xhIL-15 humanized mice.**
**(A, B)** Representative flow cytometry plots of (A) the BM and (B) the thymus of conventional NSG and hIL-7xhIL-15 KI NSG humanized mice (IL7xIL15). **(C)** Increased frequencies of CD56⁺ NK cells are detected in the BM, spleen, PB, and thymus of NSG hIL-7xhIL-15 KI humanized mice (BM, spleen, and PB: NSG *n* = 21, IL7TG *n* = 3, IL7KI *n* = 4, IL7xIL15 *n* = 17, thymus: NSG *n* = 18, IL7xIL15 *n* = 13). **(D)** Quantitative gene expression of hIL-15 by Gra1⁺Mac1⁺ mouse myeloid cells or hCD45⁺CD33⁺ human myeloid cells purified from BM of NSG (left two columns) or hIL-7xhIL-15 KI NSG (right two columns) humanized mice are presented relative to β-actin (NSG *n* = 3 and IL7xIL15 *n* = 3). **(E)** Representative flow cytometry plots of the BM, liver, and PB of hIL-7xhIL-15 KI NSG humanized mice showing CXCR6 expression among human NK cells. **(F)** Frequencies of CXCR6⁺CD56⁺ tissue-resident NK cells in the BM, liver, and PB of NSG hIL-7xhIL-15 KI humanized mice (BM *n* = 8, Liver *n* = 5, PB *n* = 4). Error bars represent mean ± SEM. *$P$ < 0.001, by two-tailed *t* test.

cytotoxicity-related genes such as *GZMA, GZMB,* and *PRF1* were up-regulated. Similarity of NK cell–associated gene expression between splenic human NK cells in hIL-7xhIL-15 KI NSG humanized mice and human PBMC-derived NK cells suggest that human IL-7 and IL-15 in the splenic microenvironment enhances human NK cell maturation. In contrast, expression of *AHR* and *IL-7R* were down-regulated in NK cells of hIL-7xhIL-15 KI NSG humanized mice and human PBMC compared with those of conventional NSG humanized mice. Among the differentially expressed genes, we analyzed the expression of KIR at the protein level by flow cytometry. Consistent with the

transcriptome analysis, expression of KIRs was higher in NK cells developed in hIL-7xhIL-15 KI NSG humanized mice compared with those in conventional NSG humanized mice (3.6 ± 0.8% in NSG, *n* = 5; 12.3 ± 0.4% in hIL-7xhIL-15, *P* < 0.001 by two-tailed *t* test; Fig S6).

# Discussion

NK cells were first reported in 1975 (Herberman et al, 1975; Kiessling et al, 1975; Sendo et al, 1975). NK cells exert potent cytotoxic function

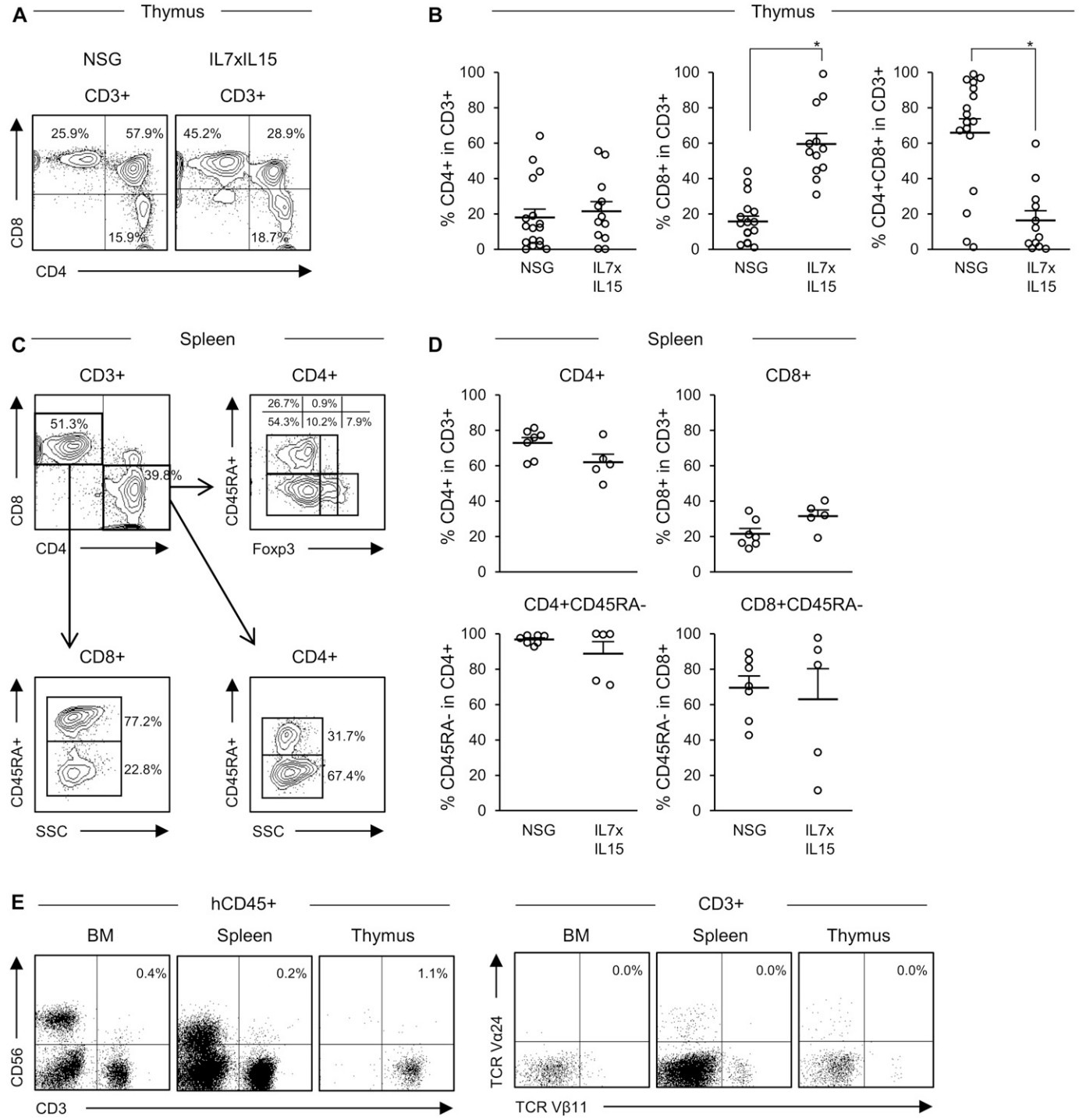

**Figure 2. Development of human T-cell subsets in NSG hL-7xhIL-15 humanized mice.**
**(A)** Representative flow cytometry plots of thymic T-cell subset of conventional NSG and hIL-7xhIL-15 KI NSG (IL7xIL15) humanized mice. **(B)** Frequencies of thymic CD4⁺ T cells, CD8⁺ T cells, and CD4⁺CD8⁺ T cells in NSG (*n* = 17) and hIL-7xhIL-15 KI NSG humanized mice (*n* = 12). **(C)** Representative flow cytometry plots of splenic T-cell subsets in a hIL-7xhIL-15 KI NSG humanized mouse. **(D)** Frequencies of splenic CD4⁺ T cells, CD8⁺ T cells, CD45RA⁻ memory CD4⁺ T cells, and CD45RA⁻ memory CD8⁺ T cells in conventional NSG (*n* = 7) and hIL-7xhIL-15 KI NSG (*n* = 5) humanized mice. **(E)** Representative flow cytometry plots showing a lack of TCR Vα24⁺Vβ11⁺ NKT cells in the BM, spleen, and thymus of a hIL-7xhIL-15 KI NSG humanized mouse. Error bars represent mean ± SEM (**P* < 0.001, by two-tailed *t* test).

against virus-infected cells and MHC-deficient tumor cells. Different from T cells, NK cells are activated when they do not receive inhibitory signals through killer cell immunoglobulin-like receptor

(KIR)–MHC interaction. This mechanism leads to rapid NK cell–mediated immune reaction and induction of apoptosis in MHC-escaping malignant cells (Anfossi et al, 2006). Like other immune

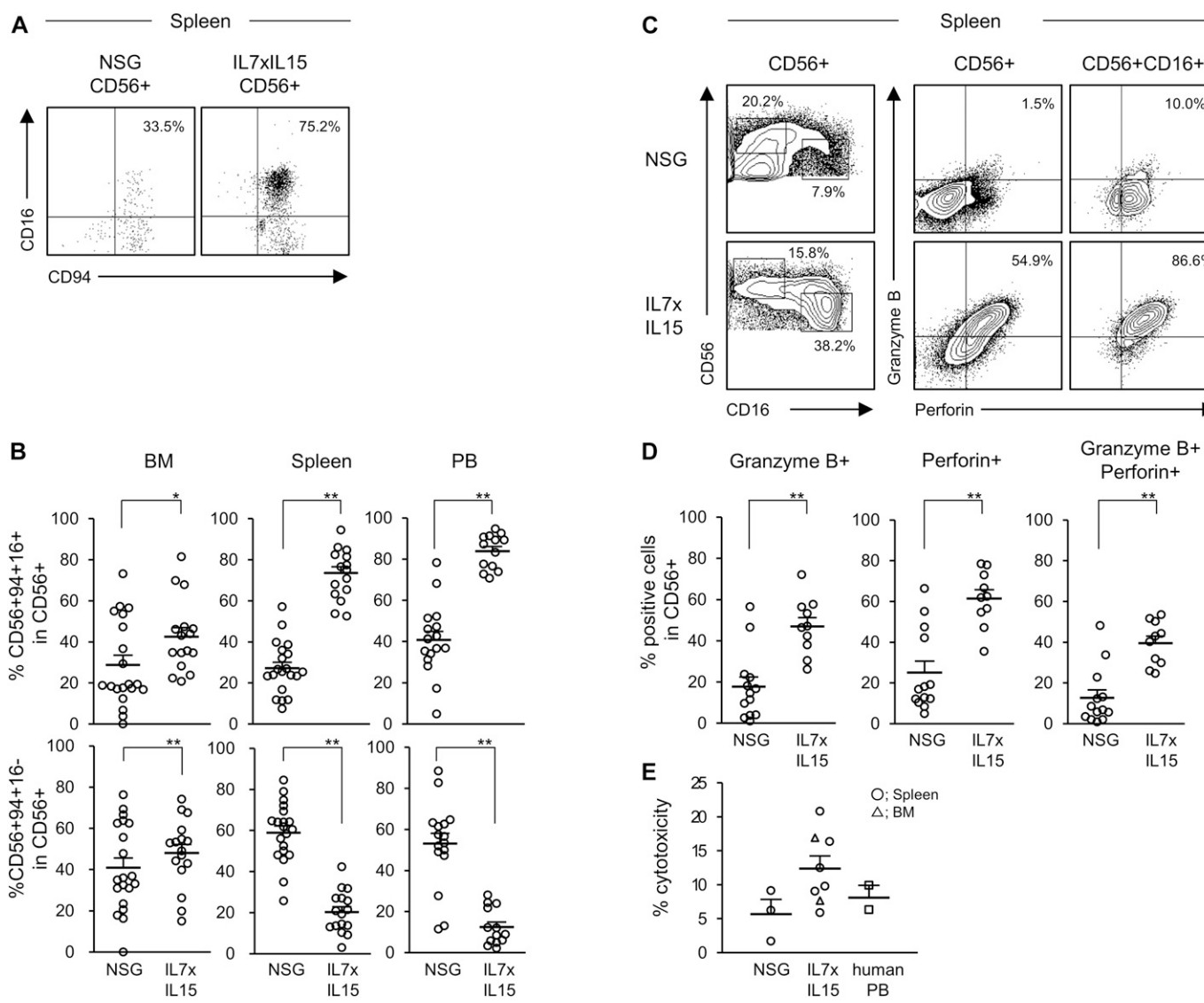

**Figure 3. NK cells show a mature phenotype and cytotoxicity in NSG hL-7xhIL-15 humanized mice.**
**(A)** Representative flow cytometry plots of splenic NK cell maturation in conventional NSG and hIL-7xhIL-15 KI NSG humanized mice (IL7xIL15). **(B)** Increased frequencies of CD56$^+$CD94$^+$CD16$^+$ mature NK cells are detected in the BM, spleen, and PB of hIL-7xhIL-15 KI NSG humanized mice (NSG: BM and spleen *n* = 20, PB *n* = 17; IL7xIL15: BM and spleen *n* = 16, PB *n* = 13). **(C)** Representative flow cytometry plots of cytoplasmic expression of granzyme B and perforin in the spleen of NSG and hIL-7xhIL-15 KI NSG humanized mice. **(D)** Frequencies of granzyme B– and perforin-positive cells among splenic CD56$^+$ NK cells in NSG (*n* = 13) and hIL-7xhIL-15 KI NSG (*n* = 10) humanized mice. **(E)** In vitro cytotoxicity of human NK cells isolated from spleen of conventional NSG, hIL-7xhIL-15 KI NSG humanized mouse spleen (circles) and BM (triangles), and from normal human PB samples (squares) against K562 are shown (NSG *n* = 3; IL7xIL15 *n* = 7; normal human PB *n* = 2). Error bars represent mean ± SEM. *P < 0.05, **P < 0.001, by two-tailed *t* test.

cells, development, maturation, and function of NK cells have been extensively studied using genetically engineered mouse models (Kennedy et al, 2000; Fehniger et al, 2001). Although mouse and human NK cell development share some aspects in their biological function and transcriptome, there are differences between human and mouse NK cells such as the use of Ly49 receptors by mouse NK cells versus the use of KIR by human NK cells for the recognition of MHC (Colucci et al, 2002). In the present study, we aimed to develop an in vivo model supporting human NK cell maturation and to assess roles of cytokine receptor signaling in human NK cell development. In particular, we aimed to address the following two questions: (1) How do human NK cells develop and distribute in primary and secondary lymphoid organs? (2) How do human NK cells become functionally mature in the microenvironment? To answer these questions, we created NSG mice expressing hIL-7 alone and those expressing both hIL-7 and hIL-15. Unlike NSG mice expressing hIL-7 alone, concurrent expression of hIL-15 resulted in enhanced in vivo human NK development in multiple organs, but not in T cells or NKT cells. Histologically, whereas human T cells and human B cells existed mainly near white pulp-like structures, most human NK cells were located outside the white pulp-like area, consistent with human splenic architecture (Witte et al, 1990). The

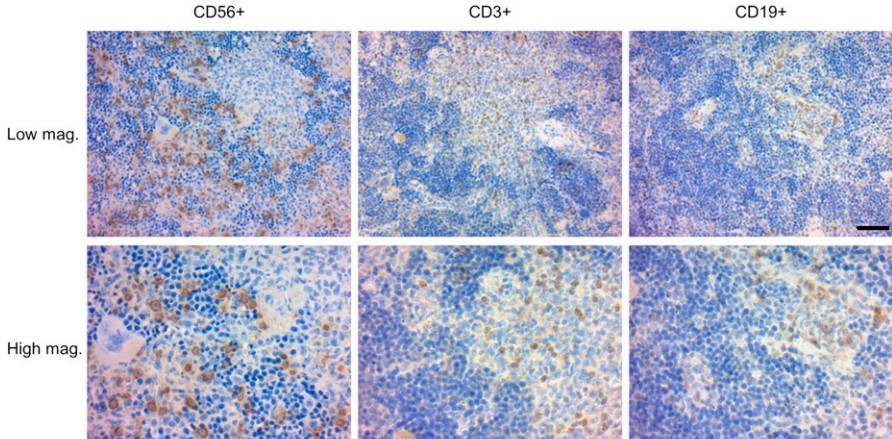

**Figure 4. Distribution of splenic T cells, B cells, and NK cells in NSG hL-7xhIL-15 humanized mice.**
Thin sections of a hIL-7xhIL-15 NSG recipient spleen stained with anti-hCD56, anti-CD3 and anti-CD19 antibodies. Low and high magnification images are shown. hCD3⁺ T cell and hCD19⁺ B cells were found within lymphoid clusters, whereas CD56⁺ NK cells are located outside the clusters. Scale bars: low magnification, 50 $\mu$m; high magnification, 20 $\mu$m.

BM is considered as a primary organ for NK cell development and maturation (Huntington et al, 2007). Although emerging evidence has suggested that thymus is also important for NK cell development, it has yet to be understood whether thymus provides essential environmental factors for mouse and human NK cells (Freud et al, 2014; Bjorkstrom et al, 2016). In our xenograft, we found CD56⁺ NK cells in both BM and thymus. Thymic NK cells were reported to express IL-7R in mice and human (Vosshenrich et al, 2006). Consistent with the report, we found IL-7R⁺CD56⁺NK cells in the thymus of hIL-7xhIL-15 KI NSG humanized mice (representative flow cytometry plots were shown in Fig S7). This result may suggest that hIL-7 signaling is important for human NK cell development particularly in thymus.

To date, the frequency of CD56⁺CD16⁺ mature human NK cells in xenotransplantation models have not fully reflected physiological levels. Our report is consistent with previous publications using in vivo administration of hIL-15 or hIL-15-hIL-15Rα complex to humanized mice (Huntington et al, 2009; Strowig et al, 2010) and more recently, using hIL-15 expressing BALBc/Rag2KO/Il2rgKO mice assessing differentiation and function of human NK cells in vivo (Herndler-Brandstetter et al, 2017). The new NSG mouse model described in the present work further supports critical roles of IL-15 in human NK cell development. In our new strain, we found CXCR6⁺CD56⁺ tissue-resident NK cells in the BM and liver. Because tissue-resident NK cells and circulating NK cells are different in cytotoxicity or cytokine production (Melsen et al, 2018), hIL-7xhIL-15 KI NSG humanized mice could be useful for studying the two distinct subsets of human NK cells. In addition to flow cytometric analysis of engrafted human NK cells, we performed transcriptome analysis by RNA sequencing. We found that the gene expression signature of human NK cells in hIL-7xhIL-15 KI NSG humanized mice was more similar to those in human PB as compared with human NK cells in conventional NSG humanized mice. It would be notable that maturation or cytotoxic markers such as KIRs, chemokine ligands (*CCL4L1* and *CCL4*), cytotoxicity-related genes (*GZMA* and *GZMB*), as well as *PRDM1* were up-regulated in the NK cells in hIL-7xhIL-15 KI NSG humanized mice because all these genes were known as NK-specific functional molecules (Fehniger et al, 1999; Parham, 2005; Yawata et al, 2006; Smith et al, 2010), and that the new humanized mice could

be a better in vivo model for assessing interaction between NK cells and diseased cells.

We also assessed potential development and survival of human NKT cells in the NSG mice expressing hIL-7 and hIL-15. In the mice engrafted with human CB HSCs, we did not see differentiation of human NKT cells in organs such as the BM, spleen, and lungs, suggesting that these two cytokines are not sufficient to support human NKT cell development. On the other hand, with in vivo transfer of human NKT cells into hIL-7xhIL-15 KI NSG mice, the injected human NKT cells were detected at higher frequencies in multiple organs as compared with NSG mice. Because NKT cells are known to activate NK cells in vivo, the model would be useful to analyze activation of human NK cells with intravenous injection of human NKT cells (Carnaud et al, 1999; Fujii et al, 2010; Yamada et al, 2016).

Together, expression of hIL-7 and hIL-15 in the NSG humanized mice resulted in efficient development of human NK cells in multiple organs in the presence of multiple human immune subsets, and the human NK cells undergo physiological maturation process and exhibit cytotoxicity. The new humanized mouse model with hIL-7 and hIL-15 expression could be used as a valuable tool for examining in vivo biology of human NK cells and for in vivo testing of NK cell–mediated cytotoxicity against cancer.

## Materials and Methods

### Human samples

Human CB cells were obtained from Tokai Cord Blood Bank and Chubu Cord Blood Bank under written informed consent. All experiments were authorized by the institutional review boards at RIKEN.

### Mice

NOD.Cg-*Prkdc^scid^Il2rg^tm1Wjl^*/SzJ (NSG) mice were raised from a breeding colony maintained at RIKEN. To generate human IL-7 transgenic mice (NSG.Cg-C57BL/6-Tg(Hu-IL7)1Rk) (hIL-7 TG), the

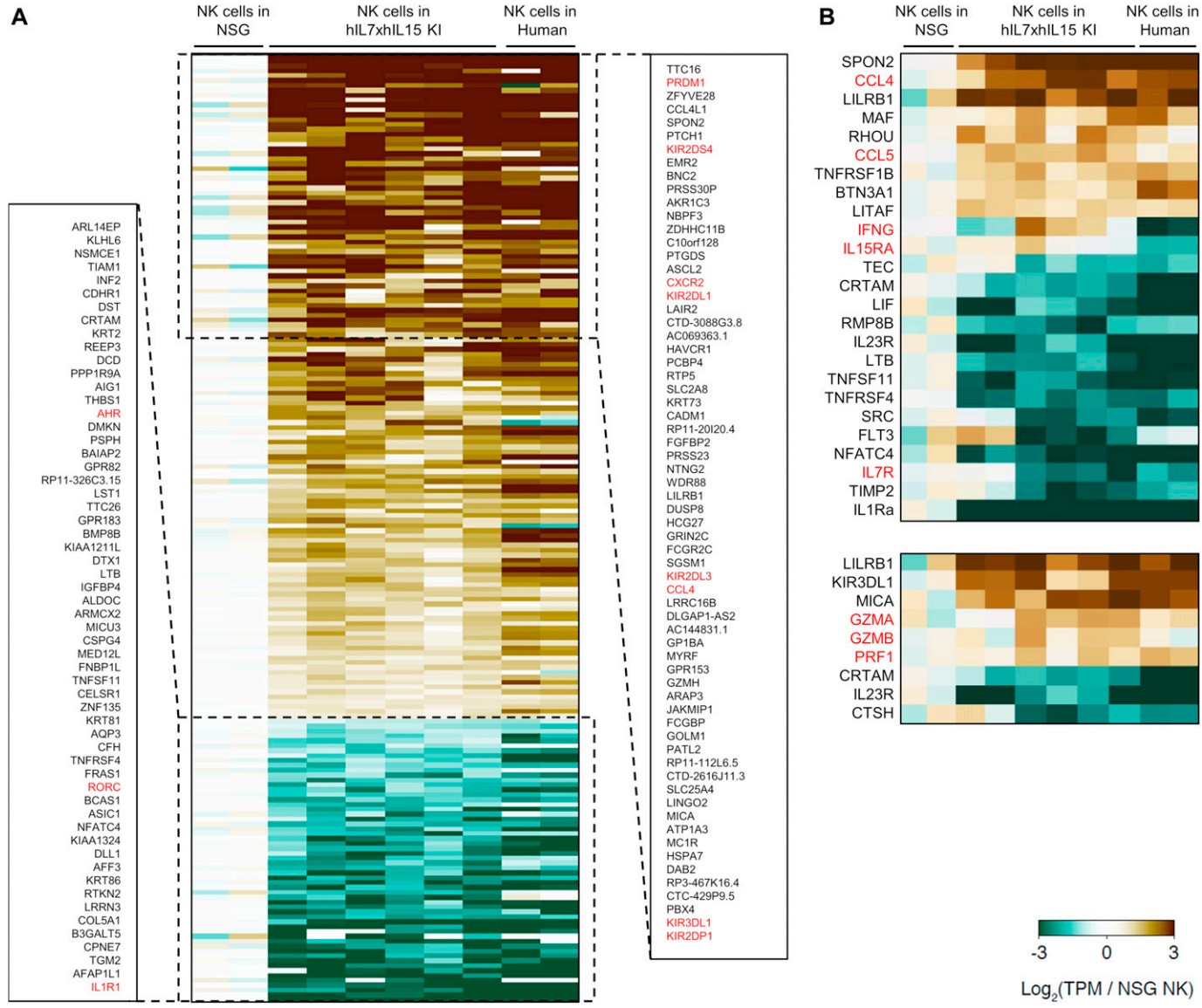

**Figure 5. Gene expression signature of NK cells in NSG hL-7xhIL-15 humanized mice.**
RNA sequencing was performed using RNA extracted from hCD56+ splenic NK cells of conventional NSG (NSG, *n* = 2) and hIL-7xhIL-15 KI NSG (hIL7xhIL15 KI NSG, *n* = 6) humanized mice. **(A)** Differentially expressing genes are shown. Gene expression profiles of human NK cells of humanized mice were compared with those from NK cells recovered from human PBMCs. **(B)** Expression of genes related to cytokine and cytotoxicity are shown.

pBACe3.6 vector containing the human IL-7 genomic region (Fig S8A; BAC clone RPCl11.C-19N15 [Thermo Fisher scientific]) was micro-injected into the pronucleus of fertilized C57BL/6J (B6/J) mouse embryos. To generate human IL-7 knockin mice (NSG.CG-STOCK-IL7$^{tm1.1(IL7)Rk}$) (hIL-7 KI mice) or human IL-15 knockin mice (NSG.CG-STOCK-IL15$^{tm1.1(IL15)Rk}$) (hIL-15 KI mice), B6/J × 129/Sv hybrid ES cells were transfected with targeting vector containing human IL-7 or human IL-15 cDNA by electroporation and cultured for 7–10 d to select homologous recombinants. Then, correctly targeted ES cell clones were aggregated with BDF2 morula to generate chimeric mice. Male chimeras were mated with B6/J females, and heterozygous offsprings were further mated with B6; SJL-Tg (ACTFLPe) 9205Dym/J mice to remove the PGK-Neo cassette (Fig S8A). hIL-7 TG mice, hIL-7 KI mice, and hIL-15 KI mice were backcrossed more than five generations onto the NSG mice. Backcrossing was performed by means of natural mating, in vitro fertilization, or round spermatid injection. Backcrossed mice at each generation were selected using a marker-assisted selection protocol for further backcrossing. Human IL-7 and human IL-15 double knockin NSG mice (hIL-7xhIL-15 KI NSG) were generated by crossing hIL-7 KI NSG mice and hIL-15 KI NSG mice. B6/J and BDF1 mice were purchased from CLEA Japan. B6; SJL-Tg (ACTFLPe)9205Dym/J mice were purchased from the Jackson Laboratory.

All mice were bred and maintained under specific pathogen free conditions at the animal facility of RIKEN Integrative Medical Sci-ences. All animal experiments were performed in accordance with

guidelines and approved by the Institutional Animal Care and Use Committee of RIKEN.

## Genotyping of NSG mice

hIL-7 TG mice, hIL-7 KI mice, and hIL-15 KI mice were genotyped by PCR (Bio-Rad, Takara Bio, or BM Equipment). Primer information used for this genotyping is shown in Fig S8B. The microsatellite markers used for marker-assisted selection protocol were selected based on sequence length polymorphisms between B6 mice and NOD mice (according to Mouse Microsatellite Date Base of Japan [MMDBJ]; https://shigen.nig.ac.jp/mouse/mmdbj/top.jsp, Table S5). All primers were purchased from Life Technologies Japan.

## Transplantation

Antihuman CD34 immune-magnetic beads (130-046-703; Miltenyi Biotec) were used to enrich human CD34$^+$ cells from CB samples. Enriched human CD34$^+$ cells were sorted for 7-AAD$^-$Lin$^-$hCD45$^+$CD34$^+$CD38$^-$CD45RA$^-$ or 7-AAD$^-$Lin$^-$hCD45$^+$CD34$^+$CD38$^-$CD90$^+$CD45RA$^-$ using FACSAria or FACSAria III (BD Biosciences). NSG, hIL-7 Tg NSG, hIL-7 KI NSG, and hIL-7xhIL-15 KI NSG newborn mice received 150 cGy total body irradiation using a $^{137}$Cs-source irradiator, followed by intravenous injection of 2.5 × 10$^3$ to 2.6 × 10$^4$ sorted HSCs. Recipient PB samples were evaluated for human hematopoietic engraftment every 4 wk starting at 5–6 wk after transplantation. Recipients with confirmed human hematopoietic engraftment at 10–24 wk after transplantation were euthanized and analyzed.

## Flow cytometric analysis

Cells obtained from PB and various organs were stained with mAbs and analyzed using FACSAria, FACSAria III, or FACSCanto II (BD Biosciences). The following mAbs were used for flow cytometry: anti-hCD3-BV421 (UCHT1), -allophycocyanin (APC) (HIT3a), and -BV786 (SK7), anti-hCD4-phycoerhthrin (PE)-Cy7 (SK3), anti-hCD8-APC-Cy7 (SK1), anti-hCD10-BV421 (HI10a), anti-hCD16-PE-Cy7 (3G8), and Pacific blue (3G8), anti-hCD19-PE-Cy7 (SJ25C1) and -BV786 (SJ25C1), anti-hCD20-APC (L27), anti-hCD33-PE (WM53), anti-hCD34-PE-Cy7 (8G12), anti-hCD38-APC (HB7), anti-hCD45-APC (HI30), -BV510 (HI30), and -APC-H7 (2D1), anti-hCD45RA-FITC (HI100), anti-hCD56-FITC (NCAM16.2), -BV421 (NCAM16.2), and -BV711 (NCAM16.2), anti-hCD90-FITC (5E10), anti-hIgD-PE (IA6-2), anti-hIgM-FITC (G20-127), anti-mCD45-APC-Cy7 (30-F11) and -BV510 (30-F11), anti-mCD11b-BV421 (Mac1) (M1/70), and Ly-6G/6C-FITC (Gra1) (RB6-8C5) (all from BD Biosciences), anti-hTCR-Vα24-PE (C15), anti-hTCR-Vβ11-APC (C21), anti-hCD127 (IL-7Ra) (R34.34) and anti-hCD158a,h (KIR2DL1/2DS1) (EB6B) (all from Beckman Coulter), anti-hCD94-PE (HP-3D9; eBioscience), and anti-hCD186-APC (CXCR6) (K041E5; BioLegend).

For analysis of regulatory T cells, cells stained with mAbs for surface antigens were fixed and permeabilized using FoxP3/Transcription Factor Staining Buffer Set (eBioscience). Permeabilized cells were stained with anti-hFoxp3-APC (clone PCH101; eBioscience) and analyzed using FACSCanto II (BD Biosciences). To analyze the expression of intracellular granzyme B and perforin, cells stained with mAbs for surface antigens were fixed and permeabilized using BD Cytofix/Cytoperm Kit (BD Biosciences). Permeabilized cells were stained with anti-human granzyme B-FITC (GB11; BD Biosciences) and anti-human perforin-PE (δG9; BD Biosciences) and analyzed using FACSCanto II (BD Biosciences).

## Measurement of plasma cytokine levels

To measure concentrations of cytokines in recipient plasma samples, PB samples were obtained from NSG, hIL-7 Tg NSG, hIL-7 KI NSG, and hIL-7xhIL-15 KI NSG mice using heparinized capillary tubes (Drummond Scientific Company). Plasma human IL-7 concentration was measured by ELISA (R&D Systems). Plasma human IL-15 concentration was measured by Bio-Plex Systems (Bio-Rad).

## Cytotoxicity analysis

Human NK cells from recipient mice and human PBMCs were enriched using NK cell isolation kit (Miltenyi Biotec). Cytotoxicity of NK cells against K562 cells were measured using LDH cytotoxicity detection kit (MK401, TaKaRa) (Korzeniewski & Callewaert, 1983; Fernandez et al, 1986; Bae & Lee, 2014). In short, 2–5 × 10$^4$ human NK cells were cultured with K562 at a ratio of 10:1 for 4 h followed by measurement of LDH in the culture supernatant by ELISA at an absorbance of 490 nm and cytotoxicity was calculated as follows: cytotoxicity (%) = [(absorbance of NK and K562 cell co-culture) − (absorbance of NK cells alone) − (absorbance of K562 alone) + (absorbance of medium alone)]/[(absorbance of K562 in 10% Triton X) − (absorbance of K562 alone)] ×100.

## Immunohistochemistry

Tissues were fixed with 4% paraformaldehyde, dehydrated, embedded in paraffin, and cut into 3-μm sections. Following deparaffinization and rehydration, antigen retrieval was performed using 0.1% calcium chloride solution (for CD56 and NCR1 [Nkp46] labeling) or Retrievagen B solution (for CD3 or CD19 labeling) (pH 9.5; BD Biosciences). Antigen retrieval was performed for 20 min at 90°C for CD56 and Nkp46 labeling and at 94°C for CD3 and CD19 labeling according to the manufacturer's protocols. Endogenous peroxidase activity and nonspecific binding were blocked with hydrogen peroxide solution and 2.5% normal horse serum, respectively. Primary antibodies used in the study were mouse antihuman CD56 (MOC-31; Abcam), rabbit antihuman Nkp46 (ab199128; Abcam), mouse antihuman CD3 (PS1; Abcam), and rabbit antihuman CD19 (EPR5906; Abcam). ImmPRESS MP-7500 was used as the secondary antibody (VECTOR). 3,3'-diaminobenzidine (DAB; SK-4100, VECTOR) and hematoxylin were used as the chromogen and counterstain, respectively. After mounting with cover glass, images were obtained using Axiovert 200 (Carl Zeiss).

## NKT cell injection

Human NKT cells and human iPS-NKT cells were prepared using a previously established method (Yamada et al, 2017). 2.0 × 10$^6$ NKT cells were intravenously injected into NSG mice and hIL-7xhIL-15 KI NSG mice. At 14 d postinjection, the recipients were euthanized and analyzed for the presence of human NKT cells in the BM and lungs.

## RNA extraction and qPCR

BM cells of NSG and hIL7xhIL-15 KI NSG humanized mice were labeled with anti-mGra1-FITC, anti-hCD33-PE, anti-mCD45-APC-Cy7, anti-Mac1-BV421, and anti-hCD45-BV510 to purify Mac1$^+$Gra1$^+$ mouse myeloid cells and hCD45$^+$CD33$^+$ human myeloid cells using FACSAria III (BD Biosciences). Total RNA was extracted using Trizol reagent (Invitrogen). Total RNA was quantified using a fluorimetric Ribo-Green assay, and RNA quality was assessed using Bioanalyzer (Agilent Technologies). Extracted RNA was converted into cDNA using Superscript III reverse transcriptase (Invitrogen). Real-time PCR reactions were performed using Platinum Quantitative PCR SuperMix-UDG (Invitrogen) on a LightCycler 480 real-time PCR system (Roche) using the following primers:

IL-15-Forward 5′-ACAGAAGCCAACTGGGTGAA-3′;
IL-15-Reverse 5′-TCCAAGAGAAAGCACTTCATTGC-3′;
IL-15-Probe 5′-(6FAM)CTTTGCAACTGGGGTGAACATCACTTTCCG (TAM)-3′.

Relative expression levels were calculated for each gene using ACTB for normalization.

### RNA sequencing analysis

Total RNA extracted from splenic CD56$^+$ NK cells in NSG recipient, and hIL-7xhIL-15 KI NSG recipient was prepared for RNA sequencing using NEBNext Ultra RNA Library Prep kit for Illumina (catalog number E7530; New England Biolabs). Final library size distribution was validated using Bioanalyzer and quantified using quantitative PCR. The DNA libraries were hybridized to a flow cell, amplified on the Illumina cBot, and subsequently run on the Hiseq 2500 (Illumina, on a 50-base single-end read mode). Raw data are deposited at the National Bioscience Database Center (accession number: hum0171). The sequence reads were mapped to the human genome (NCBI version 19) using TopHat2 version 2.0.8 and botwie2 version 2.1.0 with default parameters, and gene annotation was provided by NCBI RefSeq. The transcript abundances were estimated using Cufflinks (version 2.1.1). Cufflinks was run with the same reference annotation with TopHat2 to generate FPKM (fragments per kilobase per million mapped reads) values for known gene models.

### Quantification and statistical analysis

Quantification and statistical analysis were performed using excel. The numerical data are presented as means ± SEM. The differences were determined by two-tailed $t$ tests, and $P$ value < 0.05 was considered statistically significant.

# Supplementary Information

# Acknowledgements

We thank Chubu Cord Blood Bank for providing us with human CB and Juri Shioya for measuring cytokine concentration in the recipient plasma. This

study was supported by MEXT KAKENHI grant no. JP24111009 (to F Ishikawa), the Basic Science and Platform Technology Program for Innovative Biological Medicine from Japan Agency for Medical Research and Development (to F Ishikawa), the Research Center Network for Realization of Regenerative Medicine from Japan Agency for Medical Research and Development (to H Koseki), and CREST, Japan Science Technology Agency (to H Koseki).

## Author Contributions

M Matsuda: data curation, formal analysis, methodology, and writing—original draft, review, and editing.
R Ono: formal analysis, validation, investigation, and methodology.
T Iyoda: investigation and methodology.
T Endo: data curation, validation, and investigation.
Y Saito: conceptualization, supervision, project administration, and writing—original draft, review, and editing.
A Kaneko: data curation and formal analysis.
K Shimizu: formal analysis, validation, and investigation.
D Yamada: formal analysis, validation, investigation, and methodology.
N Ogonuki: formal analysis, investigation, and methodology.
T Watanabe: data curation, formal analysis, and validation.
M Nakayama: formal analysis, supervision, validation, and investigation.
Y Koseki: formal analysis, supervision, and investigation.
F Kezuka-Shiotani: formal analysis and investigation.
T Hasegawa: resources, supervision, investigation, and project administration.
A Ogura: conceptualization, resources, supervision, investigation, and methodology.
LD Shultz: conceptualization, supervision, and writing—original draft and project administration.
O Ohara: conceptualization, supervision, project administration, and writing—original draft, review, and editing.
M Taniguchi: conceptualization and supervision.
H Koseki: conceptualization, supervision, funding acquisition, and project administration.
S Fujii: conceptualization and supervision.
F Ishikawa: conceptualization, supervision, funding acquisition, investigation, project administration, and writing—original draft.

## Conflict of Interest Statement

The authors declare that they have no conflict of interest.

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
