## [Reviewer comments · Life Science Alliance]

Life Science Alliance

Human NK cell development in hIL-7 and hIL-15 knock-in NOD/SCID/IL2rgKO mice

Fumihiko Ishikawa, Masashi Matsuda, Rintaro Ono, Tomonori Iyoda, Takaho Endo, Yoriko Saito, Akiko Kaneko, Kanako Shimizu, Daisuke Yamada, Narumi Ogonuki, Takashi Watanabe, Manabu Nakayama, Yoko Koseki, Fuyuko Kezuka-Shiotani, Takanori Hasegawa, Atsuo Ogura, Leonard Shultz, Osamu Ohara, Masaru Taniguchi, Haruhiko Koseki, and Shin-ichiro Fujii
DOI: <https://doi.org/10.26508/lsa.201800195>

Corresponding author(s): Fumihiko Ishikawa, RIKEN; Shin-ichiro Fujii, RIEKN, IMS; and Haruhiko Koseki, RIKEN

Review Timeline:

Submission Date:	2018-09-17
Editorial Decision:	2018-10-02
Revision Received:	2019-01-29
Editorial Decision:	2019-02-11
Revision Received:	2019-02-21
Accepted:	2019-02-22

Scientific Editor: Andrea Leibfried

Transaction Report:

October 2, 2018

Re: Life Science Alliance manuscript #LSA-2018-00195-T

Dr. Fumihiko Ishikawa
RIKEN
1-7-22 Suehiro-cho, Tsurumi-ku
Yokohama, Kanagawa 230-0045
Japan

Dear Dr. Ishikawa,

Thank you for submitting your manuscript entitled "Human NK cell development in hIL-7 and hIL-15 knock-in NOD/SCID/IL2rgKO mice" to Life Science Alliance. The manuscript was assessed by expert reviewers, whose comments are appended to this letter.

As you will see, the reviewers appreciate the humanized mouse model developed. However, they also think that some more data are needed to show that the model will be useful and indeed superior to existing models. We would thus like to invite you to revise your work following the constructive suggestions made by the reviewers. Note that developing a third humanized mouse model (reviewer #2) is not required for acceptance, but perhaps you do have data at hand that compare hIL7 KI and hIL15 KI mice and thus address the issue noted by the reviewer.

Thank you for this interesting contribution to Life Science Alliance. We are looking forward to receiving your revised manuscript.

Sincerely,

- A letter addressing the reviewers' comments point by point.
- An editable version of the final text (.DOC or .DOCX) is needed for copyediting (no PDFs).
- High-resolution figure, supplementary figure and video files uploaded as individual files: See our detailed guidelines for preparing your production-ready images, <http://life-science-alliance.org/authorguide>
- Summary blurb (enter in submission system): A short text summarizing in a single sentence the study (max. 200 characters including spaces). This text is used in conjunction with the titles of papers, hence should be informative and complementary to the title and running title. It should describe the context and significance of the findings for a general readership; it should be written in the present tense and refer to the work in the third person. Author names should not be mentioned.

B. MANUSCRIPT ORGANIZATION AND FORMATTING:

Full guidelines are available on our Instructions for Authors page, <http://life-science-alliance.org/authorguide>

Reviewer #1 (Comments to the Authors (Required)):

Manuscript Nr: LSA-2018-00195-T
Matsuda et al., "Human NK cell development in hIL-7 and hIL-15 knock-in NOD/SCID/IL2rgKO mice"

The authors demonstrate that human IL-7 and IL-15 co-expression boosts NK cell development and maturation in NSG mice with human immune system reconstitution after CD34+ hematopoietic progenitor cell transfer. In addition, thymic single positive CD8+ T cell output and adoptively transferred NKT cell maintenance is better supported. They localize the increased NK cells preferentially in the red pulp in spleen, demonstrate that these contain higher frequencies of CD16, KIR, perforin and granzyme positive cells and resemble by gene expression analysis more human peripheral blood NK cells. From these data the authors conclude that the hIL-7 and hIL-15 expressing NSG mice will be a suitable tool to investigate NK cell functions and the interplay between adoptively transferred NKT cells and these cytotoxic lymphocytes.

While the presented model seems to be very interesting, its ability to develop natural cytotoxicity, namely killing of NK cell susceptible cell lines should be explored. In addition, the authors claim that NK cells and their subsets are reconstituted to higher numbers need to be substantiated with total cell numbers in addition to the so far provided cell frequencies.

Major comments:

1. The presented study demonstrates a number of interesting changes in the mononuclear cell composition of the human xenograft in hIL-7 and hIL-15 expressing mice. However, the claim that some of these are better reconstituted needs to be substantiated by total cell numbers in the investigated organs. Therefore, the authors should also report total numbers of the analyzed NK cell subsets.
2. NK cells were originally defined by function. Therefore, the authors should also explore the ability of the further matured NK cells to kill classical human NK cell targets like K562 and/or LCL721.221.
3. The liver is an organ that in humans contain high NK cell frequencies with two main subpopulations that can be distinguished by CXCR6 expression. Are these main liver NK cell subsets maintained in the hIL-7 and hIL-15 expressing mice?
4. The support for human NK cell development and maturation is assumed to be mediated mainly by trans-presentation of IL-15. Does the introduced hIL-15 become more detectable on human or even mouse myeloid cells?
5. In their gene expression data analysis the authors mainly emphasize phenotypic maturation markers. How about functional modules like cytotoxicity, cytokine transcripts and proliferation markers? These should be commented on and ideally the differential expression of the respective GO terms shown.

Minor comments:

1. The authors should briefly explain the difference between the adoptively transferred NKT and iPS-NKT cells in the Results section.
2. It would be helpful to confirm the localization of the CD56+ NK cells in the red pulp of spleen sections with another marker, like NKp46 or NKp30 or a combination thereof. At the moment the red pulp and white pulp CD56+ cells look very different. In the white pulp a lymphocyte like small cell type morphology seems to dominate, while the staining in the red pulp is preferentially myeloid cell like. An isotype control staining should be included to exclude FcR mediated red pulp staining.

In summary, the presented improved model for NK cell reconstitution should be further characterized to substantiate the authors' claim that it can serve as an improved in vivo model for human NK cells function.

Reviewer #2 (Comments to the Authors (Required)):

This article reports on the development of two new immunodeficient mouse models the NSG KI-IL7 and the double KI-IL7 and IL15 NSG. The double KI-NSG show that it will be a good recipient to study human NK cell development and also immunotherapy strategy.

The description of both model is well described. It is surprising that the KI-IL15 NSG alone was not used as a control to evaluate whether human IL15 alone also increase NK development without IL7. Indeed, it has been shown using the hSIRP-KI IL15 KI Raggamma null that these mice have a good NK reconstitution. This question whether IL7KI add extra value to the IL15. Other than adding this data I believe the comparison with NSG is appropriate.

Response to the reviewers:

Reviewer 1

The authors demonstrate that human IL-7 and IL-15 co-expression boosts NK cell development and maturation in NSG mice with human immune system reconstitution after CD34+ hematopoietic progenitor cell transfer. In addition, thymic single positive CD8+ T cell output and adoptively transferred NKT cell maintenance is better supported. They localize the increased NK cells preferentially in the red pulp in spleen, demonstrate that these contain higher frequencies of CD16, KIR, perforin and granzyme positive cells and resemble by gene expression analysis more human peripheral blood NK cells. From these data the authors conclude that the hIL-7 and hIL-15 expressing NSG mice will be a suitable tool to investigate NK cell functions and the interplay between adoptively transferred NKT cells and these cytotoxic lymphocytes.

While the presented model seems to be very interesting, its ability to develop natural cytotoxicity, namely killing of NK cell susceptible cell lines should be explored. In addition, the authors claim that NK cells and their subsets are reconstituted to higher numbers need to be substantiated with total cell numbers in addition to the so far provided cell frequencies.

Response to the Reviewers:

Thank you very much for highlighting the strength of the hIL-7xIL-15 KI NSG humanized mice. We found knock-in strategy was more useful than the transgenic approach in reproducing physiological level of human cytokines in mouse environment. We appreciate the points that you raised regarding the absolute cell number of human NK cell subsets as well as functionality of human NK cells in these mice. Please find below our response and how we specifically revised the manuscript according to your suggestions.

Major comments:

1. The presented study demonstrates a number of interesting changes in the mononuclear cell composition of the human xenograft in hIL-7 and hIL-15 expressing mice. However, the claim that some of these are better reconstituted needs to be substantiated by total cell numbers in the investigated organs. Therefore, the authors should also report total numbers of the analyzed NK cell subsets.

Response to the Reviewers:

Thank you for bringing up the importance of providing readers with absolute number of NK cells and NK cell subsets. In the revised manuscript, both percentages and absolute numbers of NK cell subsets in recipient organs are shown in Supplementary Table 3.

2. NK cells were originally defined by function. Therefore, the authors should also explore the ability of the further matured NK cells to kill classical human NK cell targets like K562 and/or LCL721.221.

Response to the Reviewers:

As per the reviewer's suggestion, we setup a functional assay in which we co-cultured NK cells and K562 and measured LDH in each well. By using controls (NK cell alone, target K562 alone),

we specifically detected lytic function of hIL-7xhIL-15 KI NSG spleen-derived NK cells (page 7, lines 30-34). The results are shown in new Figure 3E.

3. The liver is an organ that in humans contain high NK cell frequencies with two main subpopulations that can be distinguished by CXCR6 expression. Are these main liver NK cell subsets maintained in the hIL-7 and hIL-15 expressing mice?

Response to the Reviewers:

We appreciate your suggestion regarding liver NK subsets. We had not focused on this specific question in our initial submission so we performed additional experiments looking at the expression of CXCR6 in multiple recipient organs. We found higher frequencies of CXCR6+ NK cells in the BM and liver compared with peripheral blood. We now include the data in Figure 1E and F and cited a relevant reference (Vosshenrich et al., 2006).

4. The support for human NK cell development and maturation is assumed to be mediated mainly by trans-presentation of IL-15. Does the introduced hIL-15 become more detectable on human or even mouse myeloid cells?

Response to the Reviewers:

As pointed out by the reviewer, we confirmed presence of hIL-15 in the recipient plasma but had not analyzed which cells produced the cytokine in our humanized mice. Using the BM of IL-7xIL-15 humanized mice and control humanized mice, we sorted human and mouse myeloid cells followed by qPCR for hIL-15. We found expression of hIL-15 in mouse myeloid cells (page 6, lines 11-19). We have included the new data in Figure 1D.

5. In their gene expression data analysis the authors mainly emphasize phenotypic maturation markers. How about functional modules like cytotoxicity, cytokine transcripts and proliferation markers? These should be commented on and ideally the differential expression of the respective GO terms shown.

Response to the Reviewers:

As per the suggestion, we discussed with our collaborators how we should present the genomic data on functional modules. We have revised the RNAseq data and made another figure (Figure 5B) to show gene expression of ~30 transcripts by NK cells in NSG, IL-7xIL-15 NSG, and human blood.

Minor comments:

1. The authors should briefly explain the difference between the adoptively transferred NKT and iPS-NKT cells in the Results section.

Response to the Reviewers:

Thank you very much for pointing this out. We added description of NKT cells and iPS-NKT cells in Methods (page 16, lines 10-11).

2. It would be helpful to confirm the localization of the CD56+ NK cells in the red pulp of spleen sections with another marker, like NKp46 or NKp30 or a combination thereof. At the moment the red

pulp and white pulp CD56+ cells look very different. In the white pulp a lymphocyte like small cell type morphology seems to dominate, while the staining in the red pulp is preferentially myeloid cell like. An isotype control staining should be included to exclude FcR mediated red pulp staining.

Response to the Reviewers:

We agree with the reviewer that the morphology of NK cells in the red pulp is rather large and resembles that of myeloid cells. As per the reviewer's suggestion, we used FcR-blocking agent and re-tried the IHC. We also performed IHC using an antibody for NKp46. All the figures are shown below. Now, in the revised paper, we made a new Figure S4 showing NKp46 expression.

In summary, the presented improved model for NK cell reconstitution should be further characterized to substantiate the authors' claim that it can serve as an improved in vivo model for human NK cells function.

Reviewer #2 (Comments to the Authors (Required)):

This article reports on the development of two new immunodeficient mouse models the NSG KI-IL7 and the double KI-IL7 and IL15 NSG. The double KI-NSG show that it will be a good recipient to study human NK cell development and also immunotherapy strategy.

The description of both model is well described. It is surprising that the KI-IL15 NSG alone was not used as a control to evaluate whether human IL15 alone also increase NK development without IL7. Indeed, it has been shown using the hSIRP-KI IL15 KI Raggamma null that these mice have a good NK reconstitution. This question whether IL7KI add extra value to the IL15. Other than adding this data I believe the comparison with NSG is appropriate.

Response to the Reviewers:

I thank the authors for suggesting to us that we should compare IL-7xIL-15 humanized mice with IL-15 humanized mice. As per the suggestion, we have included the new data in our revised manuscript (Figure S3 and Table S4).

To summarize the data, the frequencies of human NK cells are not that different between IL-7xIL-15 double KI humanized mice and IL-15 KI humanized mice for instance in the spleen. However, we thought it important that multiple innate lymphoid cells express IL-7R and require IL-7 signaling for their functional maturation. Moreover, we found that thymic NK cells express IL-7R in our humanized mice which is consistent with a previous report. The finding is now included in Supplementary Figure 7.

Based on the results, we have made it clear in Discussion that IL-7xIL-15 double KI humanized mice might serve as a valuable research tool for investigators to examine not only function/maturation of NK cells but also interaction of human NK cells with other innate lymphoid cells and acquired immunity (page 10, lines 29-33).

February 11, 2019

RE: Life Science Alliance Manuscript #LSA-2018-00195-TR

Dr. Fumihiko Ishikawa
RIKEN
1-7-22 Suehiro-cho, Tsurumi-ku
Yokohama, Kanagawa 230-0045
Japan

Dear Dr. Ishikawa,

Thank you for submitting your revised manuscript entitled "Human NK cell development in hIL-7 and hIL-15 knock-in NOD/SCID/IL2rgKO mice". As you will see, reviewer #1 appreciates the introduced changes and only requests a minor amendment. We would thus be happy to publish your paper in Life Science Alliance pending final revisions necessary to address the remaining concern of reviewer #1 and to meet our formatting guidelines:

- please make sure that all authors are listed in our submission system and that their contributions are mentioned
- please list 10 authors et al. in your reference list
- please upload all tables as excel or word doc files
- please include accession numbers for the RNA-seq analyses

A. FINAL FILES:

-- High-resolution figure, supplementary figure and video files uploaded as individual files: See our detailed guidelines for preparing your production-ready images, <http://life-science-alliance.org/authorguide>

-- Summary blurb (enter in submission system): A short text summarizing in a single sentence the study (max. 200 characters including spaces). This text is used in conjunction with the titles of

papers, hence should be informative and complementary to the title. It should describe the context and significance of the findings for a general readership; it should be written in the present tense and refer to the work in the third person. Author names should not be mentioned.

B. MANUSCRIPT ORGANIZATION AND FORMATTING:

Full guidelines are available on our Instructions for Authors page, <http://life-science-alliance.org/authorguide>

Sincerely,

Andrea Leibfried, PhD
Executive Editor
Life Science Alliance
Meyershofstr. 1
69117 Heidelberg, Germany
t +49 6221 8891 502
e a.leibfried@life-science-alliance.org
www.life-science-alliance.org

Reviewer #1 (Comments to the Authors (Required)):

Manuscript Nr: LSA-2018-00195-TR

Matsuda et al., "Human NK cell development in hIL-7 and hIL-15 knock-in NOD/SCID/IL2rgKO mice"

The authors demonstrate that human IL-7 and IL-15 co-expression boosts NK cell development and maturation in NSG mice with human immune system reconstitution after CD34+ hematopoietic progenitor cell transfer. In addition, thymic single positive CD8+ T cell output and adoptively transferred NKT cell maintenance is better supported. They localize the increased NK cells preferentially in the red pulp in spleen, demonstrate that these contain higher frequencies of CD16, KIR, perforin and granzyme positive cells and resemble by gene expression analysis more human peripheral blood NK cells. From these data the authors conclude that the hIL-7 and hIL-15 expressing NSG mice will be a suitable tool to investigate NK cell functions and the interplay between adoptively transferred NKT cells and these cytotoxic lymphocytes.

In their revised manuscript the authors have addressed most of my concerns. They provide total numbers for the enhanced NK cell reconstitution in hIL-7 and hIL-15 expressing mice, demonstrate that the xenografted NK cells kill K562 cells similar to human peripheral blood NK cells, provide evidence also for liver NK cell engraftment and transcriptional up-regulation of cytotoxic molecules in engrafted NK cells as well as for hIL-15 in mouse myeloid cells.

Curiously, despite the higher transcript levels for granzymes A and B as well as perforin, the authors do not display the cytotoxicity of NK cells from reconstituted NSG mice alongside the NK cell cytotoxicity of their hIL-7 and hIL-15 expressing reconstituted NSG mice. This should be added to the new figure 3E in order to judge a potential functional improvement of the reconstituted NK cell compartments in the presence of hIL-7 and hIL-15.

2nd Authors' Response to Reviewers: February 21, 2019

To Reviewer 1:

We authors thank you for giving us the comments on cytotoxicity analysis. Although the number of experiments was three with human NK cells developed in non-KI NSG spleen, we added the new data to Figure 3E.

February 22, 2019

RE: Life Science Alliance Manuscript #LSA-2018-00195-TRR

Dr. Fumihiko Ishikawa
RIKEN
1-7-22 Suehiro-cho, Tsurumi-ku
Yokohama, Kanagawa 230-0045
Japan

Dear Dr. Ishikawa,

Thank you for submitting your Research Article entitled "Human NK cell development in hIL-7 and hIL-15 knock-in NOD/SCID/IL2rgKO mice". It is a pleasure to let you know that your manuscript is now accepted for publication in Life Science Alliance. Congratulations on this interesting work.

DISTRIBUTION OF MATERIALS:

Again, congratulations on a very nice paper. I hope you found the review process to be constructive and are pleased with how the manuscript was handled editorially. We look forward to future exciting submissions from your lab.

Sincerely,
